# Six-Lead Electrocardiography Enables Identification of Rhythm and Conduction Anomalies of Patients in the Telemedicine-Based, Hospital-at-Home Setting: A Prospective Validation Study

**DOI:** 10.3390/s23208464

**Published:** 2023-10-14

**Authors:** Adam Sharabi, Eli Abutbul, Eitan Grossbard, Yonatan Martsiano, Aya Berman, Reut Kassif-Lerner, Hila Hakim, Pninit Liber, Anram Zoubi, Galia Barkai, Gad Segal

**Affiliations:** 1Beyond Virtual Hospital, Sheba Medical Center, Faculty of Medicine, Tel-Aviv University, Tel Aviv 5265601, Israel; 2Faculty of Medicine, University of Nicosia, 2408 Nicosia, Cyprus; 3Dan Petah-Tikvah District at Clalit Health Services, Petah Tikva 4922297, Israel; 4Department of Pediatric Intensive Care, The Edmond and Lily Safra Children’s Hospital Sheba Medical Center, Affiliated to the Faculty of Medicine, Tel-Aviv University, Tel Aviv 5265601, Israel

**Keywords:** electrocardiography, hospital at home, arrythmia, ECG intervals, electrolyte disturbances, six-lead ECG

## Abstract

Background: The hospital-at-home (HAH) model is a viable alternative for conventional in-hospital stays worldwide. Serum electrolyte abnormalities are common in acute patients, especially in those with many comorbidities. Pathologic changes in cardiac electrophysiology pose a potential risk during HAH stays. Periodical electrocardiogram (ECG) tracing is therefore advised, but few studies have evaluated the accuracy and efficiency of compact, self-activated ECG devices in HAH settings. This study aimed to evaluate the reliability of such a device in comparison with a standard 12-lead ECG. Methods: We prospectively recruited consecutive patients admitted to the Sheba Beyond Virtual Hospital, in the HAH department, during a 3-month duration. Each patient underwent a 12-lead ECG recording using the legacy device and a consecutive recording by a compact six-lead device. Baseline patient characteristics during hospitalization were collected. The level of agreement between devices was measured by Cohen’s kappa coefficient for inter-rater reliability (Ϗ). Results: Fifty patients were included in the study (median age 80 years, IQR 14). In total, 26 (52%) had electrolyte disturbances. Abnormal D-dimer values were observed in 33 (66%) patients, and 12 (24%) patients had elevated troponin values. We found a level of 94.5% raw agreement between devices with regards to nine of the options included in the automatic read-out of the legacy device. The calculated Ϗ was 0.72, classified as a substantial consensus. The rate of raw consensus regarding the ECG intervals’ measurement (PR, RR, and QT) was 78.5%, and the calculated Ϗ was 0.42, corresponding to a moderate level of agreement. Conclusion: This is the first report to our knowledge regarding the feasibility of using a compact, six-lead ECG device in the setting of an HAH to be safe and bearing satisfying agreement level with a legacy, 12-lead ECG device, enabling quick, accessible arrythmia detection in this setting. Our findings bear a promise to the future development of telemedicine-based hospital-at-home methodology.

## 1. Introduction

Home hospitalization is already considered a viable alternative for conventional in-hospital stays worldwide. The hospital-at-home (HAH) model provides the delivery of acute care for patients suffering from several medical conditions, including community-acquired pneumonia (both bacterial and viral, as in the case of COVID-19 pneumonia), urinary tract infections (of both lower and upper urinary tracts), soft-tissue infections, and exacerbations of chronic diseases, including congestive heart failure and chronic obstructive pulmonary disease (COPD) [1,2,3,4]. Previous studies have shown not only satisfactory health outcomes in the home-hospitalization setting in the aforementioned diagnoses [5,6] but also cost reduction for the health care system, as well as increased patient satisfaction levels when compared with usual care [3,6].

Electrolyte abnormalities are common in acute patients, especially those with many background comorbidities. For example, hyponatremia is common in cancer patients, suffering from both solid tumors and hematologic malignancies, and is associated with poor clinical outcomes [7]. Hyponatremia is also common in patients presenting COPD exacerbations, associated with worse prognosis, increased risk for readmissions, and death [8]. Patients suffering from pneumonia are also prone to developing hyponatremia, mainly due to the syndrome of inappropriate andi-diuretic hormone secretion (SIADH). Here also, hyponatremia is associated with significantly worse clinical outcomes [9]. Hyperkalemia is common in end-stage renal-failure patients and is considered a life-threatening complication, associated with catastrophic anomalies of cardiac conduction [10,11]. Acute kidney injury, which is very common in acute patients [12], giving rise to myriad electrolyte imbalances, is associated with increased risk of cardiac arrythmias, beyond those directly inflicted by derangements of potassium [13]. Hypomagnesemia is also common in acutely ill patients and is associated with worse clinical outcomes in general [14] and increased risk of cardiac arrythmia in particular [15]. Hypophosphatemia, either associated with sepsis or refeeding syndrome, is also associated with increased incidence of muscle weakness, respiratory failure, lethargy, state of confusion, and arrhythmias [16,17].

In light of the above, it is reasonable to assume that the incidence of electrolyte abnormalities in the HAH setting is higher than reported. Although not included in the primary indications for HAH, the accumulated evidence regarding in-hospital patients makes it of the utmost importance to monitor electrolytes during home hospitalization as well as electrocardiographic (ECG) changes potentially arising from such anomalies. For example, potassium is an essential electrolyte used for the maintenance of cardiac cells’ resting membrane potential and its derangements might lead to recognizable changes in ECG. These changes include peaked T-waves and QRS prolongation [18]. Monitoring the dynamics of QRS prolongation, for example, is established as an essential tool for clinicians, mainly in the setting of remote monitoring [19,20]. Our basic assumption is that the incidence of electrolytes’ imbalances and resultant risk of arrythmia in the HAH setting is still unknown.

In yet unpublished research results [21], the authors appreciated the rate of electrolyte abnormalities amongst our HAH patients, reaching up to 80% of patients. During meticulous monitoring of HAH acutely ill patients, there are other, potentially life-threatening anomalies, necessitating ECG monitoring—such as those associated with covert myocardial damage during acute illness (also known as type 2 acute myocardial infarction, ascertained by increased levels of blood troponin, reaching a level of 30% in our HAH patient populations). Such findings, resembling electrolyte abnormalities, necessitate frequent ECG monitoring at patients’ homes [22].

No previous research and publications address and answer the above gap in terms of the ability to follow up potential arrythmias in the HAH population of patients. The above findings guided us to pursue an ECG technology that would enable once or twice daily ECG recording, potentially done by the patients themselves or their family members, that would be transmitted as a telemedicine input to our headquarter clinic. This would not be feasible using the legacy, 12-lead ECG machines we use in-hospital.

The KardiaMobile 6L ECG (Appendix A) is a portable ECG monitor that has shown promising results in its ability to monitor heart rate and atrioventricular (AV) conduction, as well as enable selected interval duration measurements [23]. The application of this device is easy and suitable for self-management. The output is automatically sent to the patient’s smartphone application and potentially can be also mounted to a pre-specified physician data base. In Sheba Beyond, we established an HAH service utilized for acute-care patients as an alternative for in-hospital stays of acutely ill patients. We already published previous case reports regarding the safety and feasibility of this model [24].

The aim of this study was to evaluate the reliability of the KardiaMobile portable six-lead ECG device in comparison with a standard (legacy) 12-lead ECG in the monitoring of HAH patients. The main purpose of this comparative evaluation was to provide a research-based validation for the device’s potential to identify specific arrhythmias and enable measurements of basic ECG intervals, as the means for early detection of conduction derangements, and/or myocardial damage associated with electrolyte abnormalities, and covert myocardial damage in the HAH setting.

## 2. Methods

### 2.1. Study Design

This was a prospective, comparative, open study. This study’s protocol was approved by the institutional review board of the Chaim Sheba Medical Center (IRB approval # SMC-23-3023) prior to initiation. All participants provided informed consent before enrollment in this study.

We prospectively recruited consecutive patients admitted to the Sheba Beyond Virtual Hospital, in the HAH department, during a 3-month duration. We included adult patients that were hospitalized in the HAH settings for miscellaneous clinical indications and were hemodynamically stable (systolic blood pressure above 110 mmHg, diastolic blood pressure above 50 mmHg, without new onset arrythmia upon admission, and without rapid, chronic atrial flutter or fibrillation). Exclusion criteria were age younger than 18 years, patients with acute respiratory insufficiency (patients with room-air hypoxemia below 94% were supplemented with an oxygen generator), patients with decreased or fluctuating consciousness, and those who were hemodynamically unstable. These exclusion criteria might pose a bias since they eliminate more severe patients. Nevertheless, they better represent the HAH target population.

Upon enrollment, demographic information, medical history, and baseline characteristics of the patients during hospitalization were collected in a datasheet from their electronic medical records. Data privacy was secured in accordance with requirements and regulations presented by the IRB approval. Patients’ data were anonymized.

Each patient underwent a 12-lead-ECG recording using the legacy device as part of the routine patient’s medical admission. The legacy ECG was obtained by a trained medical staff member, adhering to the standardized guidelines, at the emergency room or at an internal medicine department—prior to the patient’s transportation to their home. For patients admitted directly from their homes, the first ECG was already done at their homes. The included patients went through a subsequent additional ECG recording, by the KardiaMobile portable six-lead ECG, upon admission. The KardiaMobile ECG was also recorded by a trained medical staff member. The device was placed as per the manufacturer’s instructions to maintain consistency in the placement and quality of the recordings.

The researchers assessed the records of both ECG modalities for the following parameters: basic ECG/rhythm diagnoses, including normal sinus rhythm, sinus tachycardia (over 100 BPM), sinus bradycardia (lower than 60 BPM), atrial fibrillation, atrial flutter, first-degree AV block, second-degree AV block, complete AV block, and sinus arrhythmia. The ECG predominant intervals (PR, RR, and QT) were measured either automatically by the 12-lead legacy device or calculated from the printed output of the study device by the researchers. The comparison between devices was done using a binary classification for each parameter, being either normal or abnormal. Normal values used were 0.12–0.2 and 0.6–1.2 s for PR and RR intervals, respectively. For corrected QT (QTc), the normal values used were 0.35–0.45 and 0.36–0.46 s for men and women, respectively.

Extracted data, from patients’ electronic medical records, included laboratory results and patient characteristics. Blood levels of sodium, potassium, calcium, magnesium, troponin, and D-dimer (taken routinely, as part of our HAH service, during hospitalization on clinical grounds) were recorded, as well as the following demographics: age, gender, weight, height, main complaint, length of hospitalization, background diagnoses, medications, and clinical outcomes of the index hospitalization.

### 2.2. Statistical Analysis

Categorical variables were described as frequencies and percentages. The distribution of continuous variables was tested by a histogram and a Q-Q plot. Continuous variables that were distributed normally are presented as mean ± standard deviation, and those that were not are described as median ± IQR (inter-quartile range).

#### Assessment of the Level of Agreement of Diagnoses between Devices

We compared the degree of agreement and disagreement between the research six-lead ECG device (KardiaMobile) and the standard 12-lead ECG (legacy) device with regards to nine of the options that are included in the automatic read-out of the legacy device (normal sinus rhythm, sinus tachycardia, sinus bradycardia, atrial fibrillation, atrial flutter, first-degree AV block, second-degree AV block, complete AV block, and sinus arrhythmia). Not all diagnoses and intervals are automatically calculated and reported by the six-lead device, and some were diagnosed/calculated by the researchers, as intended by the manufacturer’s instructions.

When comparing the overall agreement between devices and methods, we translated each diagnosis and interval to a binary parameter, either normal or pathologic. We appreciated the level of consensus between devices by calculating Cohen’s kappa coefficient for inter-rater reliability (Ϗ), which subtracts the likelihood of random agreement from the overall agreement [25,26], and classifying the level of agreement (Table 1).

## 3. Results

Data regarding the ECG results of the KardiaMobile six-lead device were taken from home visits of 50 consecutive, eligible patients who were hospitalized in the Sheba Beyond HAH service between 30 April 2023 and 17 July 2023. Figure 1 shows a typical output of the research six-lead ECG as automatically generated in both the patient’s smartphone application and Sheba Beyond clinical monitoring office.

The demographic characteristics of our study’s cohort population are presented in Table 2. The data taken were of the four possible automated results provided by the device as well as researchers’ interpretations of the results for diagnoses that were not provided automatically. Additionally, the six-lead device’s intervals were manually calculated, as those data were not provided in the automatic read-out. These results were compared with the standard legacy 12-lead ECG device.

The median age of the patients was 80 years (IQR = 14), and 54% of the patients were males. A total of 14% of our patients were active smokers during their study participation. The average length of stay in an HAH was 3.42 days (not significantly different from our hospital’s average length of in-hospital stay). The baseline prevalence of cardiovascular disease was 76%, and pulmonary disease affected 22% of the patients at baseline. Additionally, 52% of the patients used antiarrhythmic drugs (which included medications from the four classes), while 20% used anticonvulsant drugs (which included the classic antiepileptic drugs as well as certain benzodiazepines, pregabalin, and gabapentin) for associated comorbidities.

Abnormal D-dimer levels were observed in 33 out of 50 patients (mostly COVID-19-positive patients), accounting for 66% of the total patient population, with a mean of 1439 ± 2021 ng/mL (normal values are up to 500 ng/mL). A total of 12 patients (24%) had abnormal troponin values (mean of 13.98 ± 31 ng/L; normal value of this high-sensitivity troponin kit is up to 12 ng/L) and were classified as suffering from occult myocardial damage, as part of a type 2 acute myocardial infarction, since none of our patients presented ST elevations nor significant ST depressions during their HAH stays.

In total, 26 (52%) patients were flagged as having electrolyte anomalies (Table 3). Among them, there were 13 cases of hyponatremia with a mean blood concentration of 137.5 ± 5.25 mg/dL (normal values are in the range of 136 to 148 mg/dL). There were no cases of hypernatremia. There were three cases of hyperkalemia (normal values are in the range of 3.5–5.2 meq/L) and three cases of hypokalemia, with a potassium mean blood concentration of 4.43 ± 0.6 mg/dL. There were nine cases of hypomagnesemia (mean blood concentration of 2.03 ± 0.23 mg/dL; normal range of values is 1.9–2.7 mg/dL) and no cases of hypermagnesemia. Additionally, there were four cases of hyperphosphatemia (mean blood concentration of 3.25 ± 0.51 mg/dl; normal range of values is 2.0–4.0 mg/dL) and no cases of hypophosphatemia. The mean blood calcium concentration was 9.072, and no cases of calcium abnormalities were observed (normal range of values is 8.1–10.4 mg/dL).

### Determining the Extent of Agreement between Devices

We assessed the level of agreement between the research six-lead device and the standard 12-lead ECG (legacy) device with regards to the nine options that are included in the automatic read-out of the legacy device. Table 4 displays the obtained results.

Regarding the level of agreement relating to ECG diagnoses, we found a level of 94.5% of cases of raw agreement (referring to the proportion of cases where there was complete agreement between the two methods) between devices when compared directly. The calculation of Cohen’s kappa for inter-rater reliability (Ϗ), which subtracts the likelihood of random agreement from the overall agreement, equaled 0.72, which is classified as substantial agreement between devices.

Regarding the level of agreement relating to ECG intervals, during comparison of the devices, agreement was measured with relation to the following intervals: PR, RR, and QT. Since QTc is derived from QT and RR, it was not added into the calculation of agreement between the devices. The overall rate of raw agreement between the devices was 78.5%, and the calculated Ϗ was 0.42, which corresponds to a moderate level of agreement.

The overall compliance of the staff members to the operation of the six-lead ECG device was very good, and all patients who gave their consent also fully complied with participation. There were no cases of consent withdrawal during this study. Also, there were no cases in which technical difficulties interrupted the completion of this study’s measurements and observations. No safety breeches were documented during this study, and no side effects were related to either the legacy or the research ECG machines.

## 4. Discussion

The potential and actual benefits of HAH are well established. Already, a 2005 Cochrane review, in which 22 trials were included, evaluating early discharge to hospital-at-home scenarios, showed that although the expected financial benefits were not met, several clinical scenarios gained a shortened length of hospital stay, so acute-care, in-hospital beds were saved [3]. Ever since, accumulated experience in HAH settings was established worldwide, with many HAH services promoting clear criteria for safe and effective management of HAH patients [27,28].

Increasing the safety belt for HAH patients is a worldwide challenge and demand. Future tasks for the HAH world include maintaining patients in their home environment without jeopardizing their health [29,30,31,32], bearing in mind that novel infrastructures and technologies, such as the IoT (internet of things), must be evaluated and adopted [33,34] for this purpose. Nevertheless, current HAH practices do not include reliable, validated means for patients’ deterioration detection and prediction [35]. Several previous publications addressed these pressing issues in several clinical settings, such as surgical and post-discharge patients [36,37]. Therefore, reliable, accessible and simple ways for arrythmia detection (amongst other physiologic parameters indicative of imminent patient deterioration) are necessary in order for the HAH world to proceed to its next steps of evolution.

A major task facing the global community of HAH is the task of moving acutely ill patients from the in-hospital environment to the HAH setting. Only such shifts will truly enable future expansion of the global HAH project and reduce in-hospital excessive morbidity and mortality. Acutely ill patients necessitate close monitoring (albeit not necessarily continuous monitoring) in order to accurately predict imminent patient deterioration and enable timely in-hospital evacuation. Current published studies on HAH patients tend to focus on financial and reimbursement issues rather than acute-patients’ safety aspects [38].

Rapid, unexpected patients’ deterioration would be, in most scenarios, a result of cardiac arrhythmias. Indeed, in-hospital preparedness focuses on clinical scenarios, such as “witnessed cardiac arrest”. While most evidence exists regarding the fate of these patients in the post-resuscitation scheme [39], such cases could be anticipated in patients suffering from myocardial damage and/or serum electrolyte anomalies and could be reflected by changes in their basic ECG rhythm and patterns [40]. Previous publications relate to older age and prolonged hospitalizations as risk factors for arrythmia development, as is the case for hospitalized COVID-19 patients [41]; nevertheless, no previous publications addressed this critical issue in HAH settings.

In the current study, aiming at future easy, accessible, and reliable means for home-based ECG recordings, we compared the outputs of two ECG devices with the intention of validating a mobile, simple-usage, six-lead ECG device. The aim of this study was to enable future routine usage of such devices in the HAH setting whenever periodic, even twice daily, ECG tracing would be appropriate—mainly for patients diagnosed as suffering from electrolyte disturbances and/or myocardial damage. We compared these devices in the true, clinical field of HAH patients. Assessment was done both directly (raw agreement rates) and by calculating the degree of agreement using the kappa coefficient. Our results show an acceptable level of agreement for the measurable parameters including major rhythm and arrythmia diagnoses and measurements of the most predominant ECG conduction intervals. It is important to emphasize that we did not record nor compare any ECG variables directly assessing potential myocardial ischemia (e.g., ST segments’ elevations or depressions and T-waves’ anomalies). Such direct assessment and diagnoses relating to the spectrum of acute coronary syndromes should rely only on full, 12-lead ECG recordings unavailable by the six-lead research device. Nevertheless, it is reasonable to assume that some levels of myocardial damage might lead to arrythmias.

The proportion of patients in this study who were predisposed to myocardial damage and arrhythmia was high: we showed that there was a substantial number of patients suffering from background diagnoses relating to the cardiovascular system and the respiratory system. Also, we found a significant portion of patients had electrolyte disturbances and/or medical treatments (either anti-arrhythmic drugs or anti-epileptic drugs), which might initiate rhythm anomalies.

The usage of the six-lead ECG as the research device in this study was performed by researchers during their clinical work in the HAH field. The overall compliance and comfort of usage was high, with no notable side effects reported.

The clinical implications of our study’s results are potentially applicable to the worldwide HAH population: increasing the safety of patients in their homes while monitoring the potential deleterious effects of occult myocardial ischemia and electrolyte abnormalities. Early diagnosis of abnormal myocardial conduction may prevent life-threatening arrythmias. Potential challenges might be an increased rate of false-positive alarms, necessitating attention and interventions by the HAH staff members. Such drawbacks should be minimized by applying future guidelines for ECG monitoring in the HAH environment.

## 5. Conclusions

Using a mobile, six-lead ECG device in the setting of an HAH was found to be safe and bearing satisfying agreement levels with a legacy, 12-lead ECG device, enabling quick, accessible arrythmia detection in this setting. This is applicable also to a high-risk population of patients, including those suffering from electrolyte anomalies and patients with increased levels of blood troponin, indicative of possible occult myocardial damage. We conclude that similar devices should be used in HAH settings, enabling the incorporation of more complex patients in this realm. The authors recommend further study of such devices used by the patients themselves. This should be investigated in a larger, prospective study. Until such prospective studies are done, we recommend all healthcare professionals engaging the HAH realm to adopt and incorporate into their routines and guidelines both serum electrolyte measurements and six-lead ECG recordings.

## 6. Study Limitations

This was a single-center study in a population of pre-defined patients (e.g., hemodynamically stable). Therefore, our findings should be related to similar-patient populations and not to diverse ethnicities. The research device was handled by the clinical staff and not by the patients themselves. Therefore, it cannot be concluded that such devices can be easily operated by the patients themselves. Future, larger studies in diverse HAH patient populations are warranted. This study was not registered in a clinical trial data base.

## Figures and Tables

**Figure 1 sensors-23-08464-f001:**
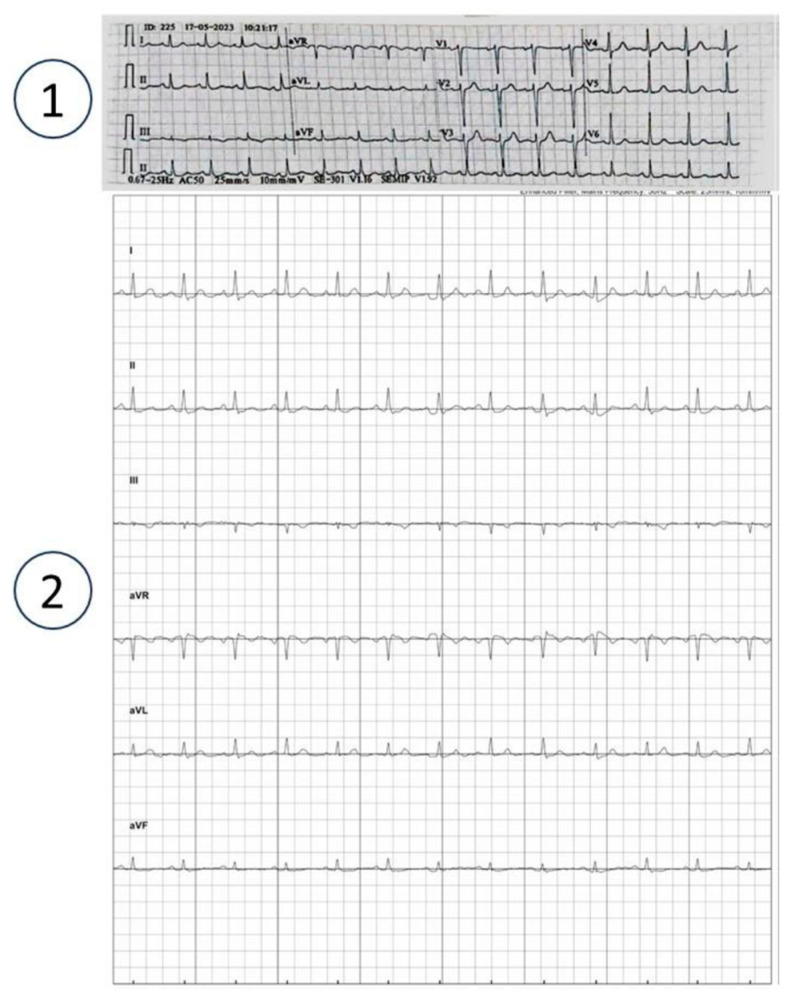
Comparison between a 6-lead ECG device’s automatic output (number 2) and that of a 12-lead for the same patient (number 1).

**Table 1 sensors-23-08464-t001:** Level of agreement according to Cohens’ kappa coefficient.

Cohen’s Kappa (Ϗ)	Interpretation
<0.00	Poor
0.00–0.20	Slight
0.21–0.40	Fair
0.41–0.60	Moderate
0.61–0.80	Substantial
0.81–1.00	Almost perfect

**Table 2 sensors-23-08464-t002:** Study’s cohort characteristics.

**Patient’s characteristics**	
Age (median, years) (IQR)	80 (14)
Male gender (%)	54%
BMI (mean)	27.28
Current smoking (%)	14
Length of stay (mean, days)	3.42 ± 0.26
**Background morbidities**	
Cardiovascular disease (%)	76
Pulmonary disease (%)	22
**Relevant background medications**	
Antiarrhythmic drugs (%)	52
Anticonvulsant drugs (%)	20

**Table 3 sensors-23-08464-t003:** Electrolytes’ blood concentrations in this study’s population.

Electrolyte	Mean ± SD Concentration (mg/dL)	Abnormally Low Levels (N)	Mean Concentration of Low Levels (mg/dL)	Abnormally High Levels (N)	Mean Concentration of High Levels (mg/dL)
Sodium	137.50 ± 5.25	13	130.23	0	N/A
Potassium	4.43 ± 0.6	3	3.26	3	5.53
Phosphorous	3.25 ± 0.51	0	N/A	4	4.35
Magnesium	2.03 ± 0.23	9	1.74	0	N/A
Calcium	9.07 ± 0.47	0	N/A	0	N/A

**Table 4 sensors-23-08464-t004:** Level of agreement between devices.

	Raw Agreement	Cohen’s Kappa (Ϗ)	Strength of Agreement
**ECG diagnosis** Normal sinus rhythm, sinus tachycardia, sinus bradycardia, atrial fibrillation, atrial flutter, 1st-degree AV block, 2nd-degree AV block, complete AV block, and sinus arrhythmia	94.5%	0.72	Substantial
**ECG intervals** PR, RR, and QT	78.5%	0.42	Moderate

## Data Availability

This study’s data will be provided upon request from the principal investigator.

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
