# Peer review of "Six-Lead Electrocardiography Enables Identification of Rhythm and Conduction Anomalies of Patients in the Telemedicine-Based, Hospital-at-Home Setting: A Prospective Validation Study"

_sensors, 2023, doi:10.3390/s23208464_

Round 1

Reviewer 1 Report

Overall, this paper explores the use of a compact, self-activated 6-lead ECG device in the Hospital-at-home (HAH) setting, comparing its performance to a standard 12-lead ECG. The study aims to evaluate the reliability and feasibility of using such a device for arrhythmia detection and monitoring in HAH patients, especially those with electrolyte abnormalities or potential cardiac issues. While the study is promising, there are several areas where comments and suggestions can be made:

1. **Title and Abstract**: The title should be more specific, summarizing the main findings. The abstract provides a good overview but could be more concise. It should clearly state the research question, methods, and key findings.

2. **Introduction**: 

   - The introduction provides a comprehensive background, but it could be more concise and focused. Try to highlight the gap in the current literature that this study addresses.

   - Clarify why this study is important and what specific problem it aims to solve. 

3. **Methods**:

   - The study design is well-described, but it could benefit from a flowchart or diagram to illustrate the patient selection and data collection process.

   - Provide more information about the Kardiamobile 6L ECG device, including its specifications and how it works, for better context.

   - Mention whether the study was registered in a clinical trials database, and if so, provide the registration number.

   - Describe any potential biases in patient selection due to the exclusion criteria.

4. **Statistical Analysis**:

   - Include more information about the statistical methods used for data analysis, especially when comparing the two ECG devices. This could involve explaining how Cohen's Kappa coefficient was calculated.

   - Consider including a CONSORT flow diagram to depict the patient enrollment and analysis process.

5. **Results**:

   - Present the results in a more structured and reader-friendly manner. Use tables and figures to summarize key findings.

   - Provide confidence intervals or standard errors for key statistics.

   - Discuss any unexpected or outlier results and possible explanations for them.

   - Clarify the significance of the high prevalence of electrolyte abnormalities and myocardial damage observed in this study, and discuss its implications for HAH.

6. **Discussion**:

   - Expand on the clinical implications of the findings. How can the use of the 6-lead ECG device benefit HAH patients, and what are the potential limitations or challenges?

   - Discuss the generalizability of the results to other healthcare settings and patient populations.

   - Consider discussing the cost-effectiveness and scalability of implementing such devices in HAH programs.

7. **Conclusion**:

   - Summarize the main findings concisely.

   - Provide clear recommendations for future research or clinical practice.

8. **Study Limitations**:

   - Emphasize any limitations of the study, such as the single-center design and the handling of the research device by clinical staff rather than patients themselves.

   - Address potential sources of bias or confounding.

9. **References**:

   - Ensure that all references are up to date and relevant to the topic.

   - Cite any relevant studies or guidelines related to the use of ECG devices in HAH settings.

10. **Language and Clarity**:

    - Proofread the paper for grammatical and typographical errors.

    - Ensure clarity in presentation, especially in the methods and results sections.

11. **Ethical Considerations**:

    - Mention any ethical considerations, such as data privacy.

12. **Visual Aids**:

    - Consider using more visual aids, such as flowcharts or diagrams, to enhance understanding.

13. **Future Directions**:

    - Suggest potential future research directions, especially in terms of larger, prospective studies involving patient-operated devices.

14. **Practical Recommendations**:

    - Conclude with practical recommendations for healthcare professionals considering the use of 6-lead ECG devices in HAH settings.

15. **Overall Structure**:

    - Ensure a logical flow in the paper, with clear transitions between sections.

The quality of English language in the paper is generally good, but there are areas where it could be improved for greater clarity and readability. Here are some specific comments on the quality of English language:

1. **Conciseness**: The paper is quite verbose in some sections. Consider condensing sentences and paragraphs to make the text more concise and easier to follow.

2. **Sentence Structure**: Some sentences are complex and may be difficult for readers to parse. Simplify sentence structure where possible to improve comprehension.

3. **Clarity**: Ensure that each sentence conveys a single, clear idea. Avoid overly long sentences that may lead to confusion.

4. **Grammar and Syntax**: The paper generally maintains proper grammar and syntax. However, it's important to review the text for any grammatical errors or awkward sentence constructions.

5. **Punctuation**: Pay attention to punctuation, including commas and semicolons, to enhance the flow of the text and clarify meaning.

6. **Redundancy**: Eliminate redundancy in the text. For example, if a point is made once, there's no need to repeat it unless for emphasis.

7. **Transition Phrases**: Use transition phrases to guide readers through the text, making it easier to follow the logical flow of your arguments.

8. **Consistency**: Ensure consistent use of terminology and formatting throughout the paper. This includes consistent capitalization, abbreviation usage, and formatting of data (e.g., tables and figures).

9. **Technical Terminology**: Define technical terms or acronyms on their first use to assist readers who may not be familiar with the specific medical terminology.

10. **Verb Tense**: Maintain consistent verb tense throughout the paper. For instance, if discussing study results, use past tense consistently.

11. **Citation Style**: Ensure that citations are in the correct format and consistent with the chosen citation style (e.g., APA, AMA).

12. **Transitions**: Improve the use of transition words and phrases between sections and paragraphs to create a smoother and more coherent narrative.

13. **Verbosity**: Be mindful of overly long and complex phrases or descriptions. Simplicity in language can enhance readability.

14. **Passive Voice**: While the use of passive voice is acceptable in scientific writing, consider using active voice when appropriate to make the text more engaging and direct.

15. **Proofreading**: Conduct a thorough proofreading pass to catch any typos or minor errors that may have been overlooked.

Author Response

On behalf of myself and all authors we thank you for your professional thorough work that improved our manuscript,

Prof. Gad Segal, MD

Reviewer 2 Report

   In this study, the authors recruited patients for 3 months, but did not mention the age of the 50 patients they recruited. At least for the research results and the entire process, we need to know their age classification.

    This study also needs to count whether any of the 50 patients have heart related problems, diabetes related problems and previous heart block problems?

    In Figure 1 author needs to add two graphs for 6-lead ECG and 12-lead ECG results, as well need to explain separately two graphs for the same patients.

    In the whole research author just mentioned the patient's smoking habits only. But for perfect calculation and research, we need to think about the patient's food habits. Based on food habits such as consumption of high fat it causes high blood pressure also.  So, we need to add those patient's food habits as well as what kind of food they had eaten before the ECG.

    On page 5, "Abnormal D-dimer levels were observed in 33 out of 50 patients", what was their physical condition and age during the research?

    Please add appropriate data related to this study, such as the patient's age, daily dietary habits, food consumption before testing, whether there have been any previous heart blockages, and whether there have been any major medical issues.

Author Response

(The authors gave the same response as above.)

Round 2

Reviewer 1 Report

The authors have addressed all my comments and questions.

Author Response

The reviewer wrote:

"The authors have addressed all my comments and questions."

Are there any other comments?

We thank you for your work!